# Acetylation of Phenylalanine Hydroxylase and Tryptophan 2,3-Dioxygenase Alters Hepatic Aromatic Amino Acid Metabolism in Weaned Piglets

**DOI:** 10.3390/metabo10040146

**Published:** 2020-04-09

**Authors:** Lu Huang, Weilei Yao, Tongxin Wang, Juan Li, Qiongyu He, Feiruo Huang

**Affiliations:** Department of Animal Nutrition and Feed Science, College of Animal Science and Technology, Huazhong Agricultural University, Wuhan 430070, China; hlldka@163.com (L.H.); yaoweilei0728@gmail.com (W.Y.); michaelwangtongxin@gmail.com (T.W.); lijuanxixi18@163.com (J.L.); heqiongyu@webmail.hzau.edu.cn (Q.H.)

**Keywords:** liver, weaned piglets, aromatic amino acids metabolism, phenylalanine hydroxylase, tryptophan 2,3-dioxygenase, acetylation

## Abstract

Weaning significantly alters hepatic aromatic amino acid (AAA) metabolism and physiological functions. However, less is known about the regulating mechanism of hepatic AAA metabolism after weaning. A total of 200 21-day-old piglets (Duroc × Landrace) were assigned randomly to the control group and the weaning group. In this study, weaning significantly decreased the concentration of phenylalanine, tryptophan, and tyrosine in piglet livers (*p* < 0.05). Additionally, through the detection of liver AAA metabolites and metabolic enzyme activity, it was observed that hepatic tryptophan catabolism was enhanced, while that of phenylalanine was weakened (*p* < 0.05). Intriguingly, acetyl-proteome profiling of liver from weaned piglets showed that weaning exacerbated the acetylation of phenylalanine hydroxylase (PAH) and the deacetylation of tryptophan 2,3-dioxygenase (TDO). Analysis of PAH and TDO acetylation in Chang liver cells showed that acetylation decreased the PAH activity, while deacetylation increased the TDO activity (*p* < 0.05). Furthermore, metabolites of AAAs and the acetylation statuses of PAH and TDO in primary hepatocytes from weaned piglets were consistent with the results *in vivo*. These findings indicated that weaning altered the PAH and TDO activity by affecting the acetylation state of the enzyme in piglets’’ livers. Lysine acetylation may be a potential regulatory mechanism for AAA metabolism in response to weaning.

## 1. Introduction

Aromatic amino acids (AAAs) (Table A1)—tyrosine (Tyr), tryptophan (Trp), and phenylalanine (Phe)—are cyclic amino acids, which contribute to the structures of proteins and are the precursors of hormones, neurotransmitters, and many other signal molecules in the body [1]. The growth development of the body and the maintenance of normal physiological functions require a stable state of AAA metabolism. The liver plays a significant role in protein synthesis and amino acid metabolism. It is also the most active organ for aromatic amino acid metabolism [2]. For decades, overwhelming consensus has emerged with countless evidence that disorders of the AAAs’ metabolism accompany various degrees of hepatic disease.

Current theories and concepts have shown that AAAs play an important role in nutritional deficiency [3]. Manary et al. reported that egg white protein is utilized better for protein synthesis after feeding AAA-rich diets to children who have kwashiorkor, when infection is most likely to be concurrent [4,5]. Additionally, Jean et al. also found that dietary supplements with AAAs could accelerate children’s net protein synthesis in nutrition rehabilitation for severe acute malnutrition, which is often accompanied by stress [5]. Reeds et al. found that infection stress increases metabolic requirements for acute phase protein synthesis, which contain a much higher content of AAAs than do either most dietary protein or muscle protein sources [6]. Therefore, it can be indicated that the metabolism of AAAs in different states is not the same, while the underlying mechanisms remain to be elucidated.

During the last decade, lysine acetylation has gradually overtaken many other major post-translational modifications (PTMs), such as ubiquitylation and phosphorylation. Lysine acetylation plays a broad role in the regulation of enzymes in metabolic process by various mechanisms [7]. Predictably, the liver, which acts as a crucial metabolic organ, is largely affected. Recently, enzymes in the liver that are related to amino acid metabolism, such as glutamate dehydrogenase (GDH), carbamoyl phosphate synthetase (CPS), aspartate aminotransferase (AST), and ornithine transcarbamylase (OTC), have been reported to be regulated by lysine acetylation [8,9,10,11], which affect the utilization of the entire amino acid. Currently, the only research in the literature regarding the acetylation of aromatic amino acid metabolic enzymes is about the acetylation of tryptophan hydroxylase 2 (TPH2), which has been reported to be involved in prenatal stress-induced depression-like behavior in male juvenile offspring rats [12]. However, the acetylation status of other metabolic enzymes of aromatic amino acids associated with stress are still largely unknown. 

Weaning is a highly stressful event as mammals experience severe nutritional deficiencies at this time [13]. Weaning stress often leads to disorders in the immune system and impaired digestive function [14,15]. Weaning can lead to an increased demand for essential amino acid (e.g., AAAs) from protein breakdown and dietary protein intake to synthesize negative and positive acute-phase proteins (APPs) [16]. The present objective is to unravel the regulation mechanisms of hepatic aromatic amino acid metabolism by protein lysine acetylation in weaned piglets and to find a novel propitious method to prevent weaning stress in domestic animals and in human beings.

## 2. Results

### 2.1. Weaning Stress Reduces the Content of Hepatic Aromatic Amino Acids in Piglets

It is well known that weaning leads to remarkable stress in piglets. The plasma tumor necrosis factor-α (TNF-α), interleukin 6 (IL-6), and cortisol concentrations were monitored. The concentrations of IL-6, TNF-α, and cortisol reached a peak value at day 1 and gradually decreased (Figure 1A–C), which indicated that the pigs were subjected to weaning stress. To examine the amino acid metabolism in the liver of weaned piglets, hepatic free amino acid contents were tested. Hepatic-free amino acid profiles of piglets are shown in Table 1. The glutamic acid (Glu) and alanine (Ala) concentrations were very high (2340.47 nmol/mL, 2540 nmol/mL) in weaned piglets. The concentration of lysine (Lys) was greater (*p* < 0.05) in weaning group than that of the controls. However, the weaning group had a lower (*p* < 0.05) hepatic concentration of aromatic amino acids (Tyr, Phe, and Trp) compared with that of control group. These results suggested that amino acid utilization has been redistributed in response to weaning stress. A total of 5188 proteins were picked out from 36,757 peptides in 20 samples (*n* = 10 for each group). A number of 203 differentially expressed proteins from the weaning and control groups are shown in Appendix A. Identified differentially expressed proteins were shown by volcano plot, which demonstrated 53 proteins with lower expression levels and 150 proteins with higher expression levels in the weaning group (Figure 1D). Significant signaling pathway was found in Kyoto Encyclopedia of Genes and Genomes (KEGG) analysis. Pathways mainly related to immune response, amino acid metabolism, and inflammatory response were enriched (Figure 1E). Then, the levels of hepatic acute-phase proteins were measured, which contained large amounts of aromatic amino acids. The levels of C-reactive protein (CRP), pig-major acute phase protein (Pig-MAP), haptoglobin (HP), and serum amyloid A (SAA) in liver were significantly greater (*p* < 0.05, Figure 1F–J). The amino acid composition of the acute phase protein is high in aromatic amino acids. Results shows that AAAs are more commonly used to synthesize acute phase proteins under weaning stress.

### 2.2. Weaning Stress Inhibits Phenylalanine and Tyrosine Catabolism and Promotes Hepatic Tryptophan Catabolism

The conversion of phenylalanine to tyrosine is irreversible in the liver by the action of phenylalanine hydroxylase (PAH), which is a precursor of catecholamine synthesis. A small portion of tryptophan produces serotonin by tryptophan hydroxylase, and approximately 95% tryptophan is further metabolized by forming kynurenine (Kyn) by the action of tryptophan 2,3-dioxygenase (TDO). Therefore, metabolites and metabolic key enzymes of AAAs in the piglet liver were detected for further analysis of hepatic AAA metabolism. The results showed that weaning stress increased the TDO activity (*p* < 0.05, Figure 2A). Congruous with the increased enzyme activity, the levels of Kyn concentration in liver and plasma were increased significantly (*p* < 0.05, Figure 2B,C). The levels of the plasma 5-hydroxytryptamine (5-HT) concentration were decreased in weaned piglets (*p* < 0.05, Figure 2D). The activity of PAH, which catalyzes the irreversible conversion of phenylalanine into tyrosine, was found to decrease in the livers of weaned piglets (*p* < 0.05, Figure 2E). In addition, tyrosine, a metabolite of phenylalanine, was decreased in the weaned piglets significantly (*p* < 0.05, Figure 2F). As expected, the concentrations of tyrosine-related metabolites, such as 4-hydroxyphenylpyruvate (HPPA) and homogentisic acid (HGA) were significantly decreased in the liver from weaning group (*p* < 0.05, Figure 2G,H). To further test metabolism of tryptophan and phenylalanine directly, the weaned piglet hepatocytes were treated with [U-^13^C] tryptophan and [U-^13^C] phenylalanine, respectively. The results showed that the Kyn labeled with [U-^13^C] was nearly 90% of the total, but the tyrosine labeled with [U-^13^C] was merely ~10% (Figure 2I). Predictably, these results suggested that the catabolism of phenylalanine was inhibited, and the catabolism of tryptophan was enhanced in the liver of weaned piglets.

### 2.3. Lysine Acetylation Modifies the Key Enzyme of Hepatic AAAs Metabolism, thus Altering the Utilization of AAAs

Detected acetylated peptides in the liver of weaned piglets were assessed to get further insight into the effects of weaning in the liver acetylome. To determine acetylated hepatic proteins and their modification sites, a proteomic method was applied by high-resolution liquid chromatography tandem mass spectrometry (LC-MS/MS). The false discovery rate (FDR) of the identified acetyl peptides was 1%. The sum of the differential quantification proteins and sites were summarized (Figure 3). The mass spectrometry (MS) data validation was shown in supporting material (Appendix A). The mass errors of all these identified peptides were verified, with a distribution that was approximately zero, and most <0.02 Da, which indicated that the results were satisfactory. In addition, most peptides were 8–20 tryptic peptides in length, which indicated that the preparation for all the samples met the necessary standards. Proteins are involved in AAAs’ metabolism. Acetylation of six enzymes of AAAs metabolism in the liver was affected by post-weaning. Specifically, the acetylation level of the TDO were reduced in the liver of weaned pigs, compared with that of control group, while the acetylation level of PAH was upregulated in weaned piglets. Both PAH and TDO had different levels of acetylation in the livers of weaned piglets, and their acetylated lysine residues were identified (PAH: K38, K204, K282; TDO: K94, K212), which were predicted to influence the enzyme binding or activity (Table 2).

### 2.4. Acetylation of PAH and TDO Modulates Its Activity

Results show that weaning decreased the activity of PAH, while the activity of TDO increased significantly compared to that of the controls, which was congruous with observations of previous studies. To confirm that the activity of these two enzymes could be influenced by acetylation/deacetylation, activity of ectopically expressed and endogenous PAH and TDO was detected in Chang liver cells under the treatment of trichostatinA (TSA) and nicotinamide (NAM). TSA and NAM indeed enhanced acetylation levels of ectopically expressed PAH and TDO by inhibiting deacetylation (Figure 4A). TSA and NAM decreased the activity of endogenous PAH and TDO significantly (*p* < 0.05, Figure 4B,C). Consistently, the activity of the wild-type (WT) PAH was increased after being treated with TSA and NAM in transfected HEK293T cells, but not the PAH mutant with three acetylated lysine residues replaced by arginine. The activity of WT TDO was decreased, while the activity of TDO mutant with two acetylated lysine residues replaced by arginine was not significantly different (Figure 4D,E). These observations suggested that acetylation of PAH and TDO altered the enzyme activity, as a result of the previously identified observation of the acetylation of PAH at K38, K204, and K282 and the deacetylation of TDO at K94, K212. Furthermore, the deacetylation of immunopurified PAH and TDO *in vitro* by CobB, a known Sir2-like bacterial lysine deacetylase, increased the PAH and TDO activity, which indicated that acetylation directly activates PAH and TDO (Figure 4F,G). 

### 2.5. Acetylation of PAH Inhibiting Phenylalanine Catabolism and Deacetylation of TDO Promote Tryptophan Catabolism in Primary Hepatocytes

We used a weaned piglet model to study the metabolic consequences of expressing PAH and TDO acetylation mutants in the liver. A cellular experiment was performed to determine whether acetylation of PAH and TDO could affect AAA metabolism dealing with weaning stress. Primary porcine hepatocytes were isolated from the liver of unweaned piglets (control group) and weaned piglets (weaning group). The metabolites of aromatic amino acids, the activity of metabolic enzymes, and the state of acetylation were examined. The hepatic Kyn content increased significantly (*p* < 0.05, Figure 5A) in weaning group compared with that of unweaned piglets; however, the 5-HT, HPPA and HGA content significantly decreased (*p* < 0.05, Figure 5B–D). Of note, the acetylation levels of PAH and TDO were ascertained by measuring in the cells of the control and weaning groups. Interestingly, the TDO acetylation level decreased, and the PAH acetylation level increased significantly (*p* < 0.05, Figure 5E–H). In addition, the TDO activity increased by 18% in weaned piglets compared with that of controls, while the PAH activity significantly decreased (*p* < 0.05, Figure 5I,J). Then, primary hepatocytes from unweaned pigs were infected with a recombinant adenovirus that uses the K-to-R mutation of both PAH and TDO. As expected, the experimental results were congruous with those of the control group.

## 3. Discussion

Weaning is an abruptly tremendous stress in the lives of newborns, and it can lead to stunting and susceptibility to diseases in mammals. This issue is particularly serious during the first week after weaning in commercial swine husbandry, which often leads to “postweaning stress syndrome” (PWSD) [17]. Weaning stress causes the absorption and metabolism of nutrients in various organs of the body to be restricted [18]. Remarkably, the liver is a vital organ, which plays an important metabolic role in regulating the homeostasis of amino acids, lipids, and glucose during changes of nutrient availability and cellular metabolic status [19,20]. In particular, amino acids will be redistributed and used primarily for hepatic gluconeogenesis, acute-phase protein, and other important compounds with the immune function in response to stress [21,22]. 

In the present study, weaned piglets were used to imitate a weaning stress model. Weaning causes stress, and aromatic amino acids (e.g., Phe) are used to synthesize acute phase proteins [6]. It was observed that the concentrations of glutamic acid (Glu), glycine (Gly), aspartic acid (Asp), and alanine (Ala) were significantly increased in the liver of weaned piglets. As is well known, amino acids in the liver shift from being used for growth to prioritize metabolic changes in the immune system in response to stress, which means that the amino acid demand pattern of the animal under stress changes [23,24]. To provide free amino acids for liver utilization, on the one hand, the capacity of hepatic protein catabolism is reinforced in the state of stress. On the other hand, under the action of amino acid transaminase, the intermediate metabolites Glu, Gly, Asp, and Ala will be produced in large quantities, which indicates that the liver is in an extremely active state. Nevertheless, amino acids that are produced by the catabolism of liver proteins cannot satisfy the synthesis of APPs, which cause amino acids to be produced by the catabolism of muscle proteins in the liver [6]. This is also a primary reason for the observed increase in the concentration of hepatic free amino acids. However, the hepatic free amino acids and the amino acids catabolized by muscle protein in the liver still cannot satisfy the synthesis of APPs during the weaning stress period. In our study, it was observed that the concentration of Phe, Trp and Tyr were reduced in the liver from weaning group. These results evidenced that the utilization of hepatic amino acids was redistributed, while demand for AAAs was reinforced under weaning stress.

Phenylalanine is a basic and indispensable amino acid, which cannot be synthesized by the body [25]. Under normal physiological conditions, phenylalanine is largely metabolized in the liver and in other tissues to produce tyrosine, and then in the nervous system and adrenal medulla, synthesis of certain neurotransmitters and hormones, such as epinephrine, norepinephrine, dopamine, and melanin, occurs [26]. Many factors, such as liver disease, genetics, immunity, and neurohumor, can cause dysfunction of phenylalanine metabolism, which results in the obstruction of the conversion of phenylalanine into tyrosine [27,28,29]. The liver is the main organ where phenylalanine is metabolized by phenylalanine hydroxylase (PAH). Stress leads to the destruction of PAH in the liver, which results in the dysregulation of phenylalanine metabolism, but the mechanism is unclear. Indeed, weaning stress resulted in a decrease in PAH activity in the current study. As expected, the hepatic tyrosine content also decreased in weaned piglets. At the same time, the production of 4-hydroxyphenylpyruvate and homogentisic acid was inadequate due to the decrease of tyrosine sources in the liver. However, level of phenylalanine in the blood did not reach a high level. A reasonable explanation was that phenylalanine was used extensively for the synthesis of APPs without accumulation in the body, even under the low activity of the PAH enzyme. Furthermore, this study used ^13^C-labeled phenylalanine to determine which is the major pathway of phenylalanine metabolism during the response of weaning stress. The data found that the percentage of ^13^C-labeled APPs was considerably higher than that of tyrosine. Therefore, these results indicated that although weaning stress resulted in a decrease of PAH activity, phenylalanine was mainly consumed in the form of synthetic APPs.

Tryptophan plays a crucial role in regulating growth, mood, behavior, and immune response [30]. There are two main metabolic pathways for tryptophan in the body, which are oxidative decarboxylation to synthesize 5-HT and the catabolism of kynurenine to produce CO_2_ and H_2_O. Previous studies suggested that 5-HT could reduce stress and inflammation and significantly improve mood disorders. This experiment found that weaning stress significantly reduced serum tryptophan and 5-HT concentration. The precursor of 5-HT biosynthesis is tryptophan, and its synthesis rate is directly regulated by tryptophan [31,32,33]. However, the hepatic tryptophan metabolite Kyn, the Kyn/Trp ratio, and TDO activity were significantly increased. TDO was the rate-limiting enzyme for the metabolism of Trp to Kyn. Under physiological conditions, liver TDO is the main tryptophan oxidase and plays a vital role in regulating the plasma tryptophan content. Due to the high activity of TDO, the catabolism of tryptophan was accelerated, which resulted in a decrease in the 5-HT content. Increased hepatic TDO activity in weaning group led to decreased plasma tryptophan and an increased Kyn/Trp ratio, which was consistent with the result that stress induced an increased TDO activity and increased tryptophan metabolism in the human liver. In this study, the increased ratio of Kyn/Trp was the direct result of the decrease in the serum tryptophan concentration caused by weaning stress and the increase in the kynurenine concentration, which indicated that the body’s tryptophan catabolism was enhanced. This is consistent with reports that immune stress-enhanced human tryptophan metabolism and the Kyn/Trp ratio [34]. Thus, weaning stress leads to increased TDO activity, which inhibits the metabolism of tryptophan to 5-HT and promotes the conversion of tryptophan to Kyn in the liver.

A previous theory postulates that hepatic amino acid metabolism is regulated by lysine acetylation, which was proven to be a prominent posttranslational modification. Compared to other PTMs, lysine acetylation is a widespread modification in enzymes that catalyze intermediate metabolism in fatty acid metabolism and the tricarboxylic acid (TCA) cycle [35,36,37]. In this research, two major hepatic enzymes in the AAA pathways, PAH and TDO have been found to be significantly changed in their acetylation level, which suggested that AAAs’ metabolism may be altered by lysine acetylation under weaning stress. In addition, our previous study suggested that acetylation plays a major regulatory role in reprogramming mitochondria to amino acid consumption under dietary stress, which was similar to its role in weaning [11]. Intriguingly, the results showed that weaning increased the PAH acetylation level, while it decreased the TDO acetylation level in piglet liver. Furthermore, by using a tandem mass spectrometry (MS) analysis, three specific acetylation sites (K38, K204, and K282) of PAH and two specific acetylation sites (K94 and K212) of TDO were identified. To investigate whether the PAH and TDO activity are modified by those sites, the endogenous and activity of ectopically expressed PAH and TDO was detected in Chang liver cells after treatment with NAM and TSA. Consistently, these acetylated lysine residues in transfected HEK293T cells were replaced with arginine. Of note, treatment with NAM and TSA significantly decreased PAH and TDO activities at their acetylated lysine residues. As NAM is an inhibitor of the deacetylase Sirtuin 1 (SIRT1), it seems that SIRT1 plays an important role in aromatic amino acid metabolism. Based on previous studies, the activity of mammalian PAH was tightly controlled by three main mechanisms in the liver: the concentration of the natural cofactor (6R)-BH_4_, L-Phe, and phosphorylation [38,39,40]. In this case, lysine acetylation may act in other regulation mechanisms of PAH. In addition, how TDO activity is regulated has been poorly studied; these results may also provide new insight into hepatic Trp metabolic regulating mechanisms.

The metabolic and physiological consequences of PAH acetylation and TDO deacetylation were studied in primary porcine hepatocytes isolated from the liver of weaned piglets. These results found that the acetylation of PAH increased, while the activity of PAH and the metabolites of phenylalanine decreased. In contrast, the TDO acetylation level decreased, while the activity of TDO and the metabolites of tryptophan increased. These results confirmed that the acetylation levels of PAH and TDO were directly related to weaning, which affected AAAs’ metabolism in turn. To further demonstrate the effects of PAH and TDO acetylation on AAAs metabolism in liver, primary hepatocytes from unweaned pigs were infected with recombinant adenovirus, using the K-to-R mutation of PAH and TDO, which abolished the positive charge and acted as a substitute for acetylation [41]. As expected, the mutation of PAH inhibited phenylalanine metabolism, and the mutation of TDO inhibited tryptophan metabolism. These results suggested that dynamic acetylation and the deacetylation of proteins has pleiotropic effects on cell metabolism. There is no doubt that acetylation/deacetylation may play roles in cellular processes that are independent of transcription. These observations showed that weaning altered the acetylation of PAH and TDO, which affected the metabolism of aromatic amino acids in hepatocyte. It should be considered that an acetylated status is a comprehensive result of acetyltransferase and deacetylase in specific cells and even organisms.

In summary, the results that are presented in the current study demonstrate that acetylation of PAH and the deacetylation of TDO are essential for the regulation of hepatic metabolism of aromatic amino acids after weaning *in vivo*. Here, we report the first investigation of lysine acetylation from a PAH and TDO perspective in the liver of weaned piglets. That is, lysine acetylation may be a potential regulation mechanism of the metabolism of aromatic amino acids, and the acetylation/deacetylation of PAH and TDO is closely related to the metabolism of aromatic amino acids by changing the activity of metabolic enzymes. Acetyltransferase and deacetylase determine the acetylation status in the body by comodulating the intracellular protein acetylation/deacetylation may act as a central role in regulating intermediate metabolism and metabolism-related diseases. The next step is to find the acetyltransferase for PAH and deacetyltransferase for TDO, which will be important in studying the regulatory targets of AAA metabolism and provide novel therapeutic strategies.

## 4. Methods and Materials

### 4.1. Experimental Design 

According to the protocols for animal care and use of laboratory animals under the approval of Huazhong Agricultural University Institutional Animal Care and Use Committee, the experiments were conducted. A total of 200 7-day-old piglets (Duroc × Landrace) from 20 litters were divided into two groups by litter: the normal suckling control group (*n* = 100) and the weaning group (*n* = 100). From day 7 to day 20, all piglets were placed in gestation crates of the same farrowing pens with their sows. From day 21 to day 28, the controls remained with their sows, while piglets of the weaning group were weaned and moved into nursery pens. All piglets were given free access to distilled water and feed, and the temperature in the nursery pens was approximately 30 °C, with a relative humidity of 50%–70%. Pens were cleaned regularly to remove feces, and the barn was well- ventilated.

### 4.2. Sample Collection

At day 20 after birth (day 0 after weaning), 10 piglets from 10 different litters with similar body weights were randomly picked out and sacrificed (half females and half males) before dividing the piglets into two groups. At day 1 to day 7 after weaning, piglets of similar body weights were selected, one per litter. Piglets selected were anaesthetized by intramuscular injection of sodium pentobarbital (40 mg/kg BW) (Merck, Darmstadt, Germany). Venous blood sample was received before slaughter. Blood samples were placed on ice at once, and were subsequently centrifuged at 800× *g* and 4 °C for 16 min and plasma was stored in 200 μL aliquots at −80 °C. The blood on the left lobe of the liver was rinsed quickly using saline. After rinsing, the liver sample was placed in liquid nitrogen immediately and stored at −80 °C. 

As we previously described, liver samples from unweaned piglets and weaned piglets were prepared for protein extraction [42]. In brief, 50 mg frozen liver samples were pulverized in liquid nitrogen with pestle and mortar. These powders were dissolved in 400 μL lysis buffer (7 M solid urea, 4% CHAPS, 15 mM Tris and 2 M thiourea, pH 8.5) adding 1% protease inhibitor (100 ×) (GE Healthcare, Piscataway, NJ, USA). Put on ice, and mix 3 times. An Ultrasonicator Model VCX 500 (Sonics & Materials, Newtown, CT, USA) was run for 10 min to decompose the mixture at 0 °C at 20% power output under a cycle of 2 s on and 8 s off. To completely rupture the cell membranes, the lysed cell suspension was added a supplement of 1% (*v*/*v*) nuclease mix (100×) (GE Healthcare, Boston, MA, USA) and placed at room temperature for 1 h, and subjected to sonication as described above [43]. Subsequently, centrifuge the homogenate at 13,000× *g* and 4 °C for 10 min. By the Bradford method, the concentrations of proteins were determined. The protein aliquots were stored at −80 °C.

### 4.3. Protein Preparation

The liver samples were ground and then sonicated on ice 3 times by a high-intensity sonicator (Scientz, Ningbo, China) in lysis buffer (3 µM TSA, 50 mM NAM, 0.1% protease inhibitor cocktail, 10 mM DTT, 8 M urea). Centrifugation was performed for 10 min at 20,000× *g* and 4 °C to remove residual debris. Subsequently, the protein was deposited and the supernatant was discarded by centrifugation. The precipitate was washed and re-dissolved in buffer (100 mM TEAB, 8 M urea, pH 8.0). The concentration of protein was determined by the 2-D Quant kit. The protein solution was reduced by 10 mM DTT and alkylated by 20 mM IAA (Sigma, St. Louis, MO, USA) for digestion. An amount of 100 mM TEAB was added to less than 2 M in urea concentration to dilute the protein samples. Trypsin (Promega) was added for the first digestion with a 1:50 mass ratio of trypsin-to-protein and with a 1:100 mass ratio for the second 4-hour-digestion. The iTRAQ labeling and analysis were implemented by Novogene Bioinformatics Technology Co. Ltd. (Beijing, China). Peptides were desalted by Strata × C18 SPE column (Phenomenex, Torrance, CA, USA) and vacuum-dried after trypsin digestion. Peptides were reconstituted in 0.5 M TEAB and processed by a 6-plex TMT kit (Thermo, Waltham, MA, USA). The mixtures were subsequently incubated at room temperature for 2 h and combined, desalted, and lyophilized.

### 4.4. Biochemical Analyses 

1% and 10% liver homogenates were manually prepared using a glass tissue grinder with 0.86% ice-cold saline as medium. Hepatic-free AA concentration was defined by high-performance liquid chromatography (HPLC) [44]. Two other methods were used to analyze cysteine and proline [45,46]. The plasma concentrations of cortisol, tumor necrosis factor-α (TNF-α), and interleukin 6 (IL-6) were measured using ELISA (QIAGEN, Dusseldorf, Germany) kits.

### 4.5. Affinity Enrichment

Dissolve tryptic peptides in NETN buffer (100 mM NaCl, 0.5% NP-40, 50 mM Tris-HCl, 1 mM EDTA, pH 8.0) (Sigma, St. Louis, MO, USA), and subsequently incubated with gentle shaking at 4 °C overnight with antibody beads (PTM Biolabs, Chicago, IL, USA), which were washed 4 times with NETN buffer and twice with ddH_2_O. An amount of 0.1% TFA (Sigma, St. Louis, MO, USA) was used to elute the bound peptides. The eluted fractions were combined, vacuum-dried, and rinsed by C18 ZipTips (Millipore, Burlington, MA, USA), and subsequently subjected to LC-MS/MS analysis.

### 4.6. LC-MS/MS Analysis

Peptides were dissolved in 0.1% FA (Fluka), and loaded directly onto a reversed-phase precolumn (Acclaim PepMap 100, Thermo, Waltham, MA, USA). A Q Exactive^TM^ hybrid quadrupole-Orbitrap mass spectrometer (Thermo, Waltham, MA, USA) was used to analyze the resulting peptides. The peptides with a treatment of NanoSpray Ionization (NSI) source was analyzed by MS/MS in Q Exactive^TM^ (Thermo, Waltham, MA, USA). At a resolution of 70,000, complete peptides were detected in Orbitrap. For MS/MS, peptides were picked up by a normalized collision energy (NCE) setting of 28. At a resolution of 17,500, Orbitrap was used to detect the ion fragments. For the top 20 precursor ions above the threshold ion count of 5E3, a data-dependent process was performed between 1 MS scan and subsequent 20 MS/MS scans alternately. The electrospray voltage applied was 2.0 kV. To prevent Orbitrap from overfilling, the automatic gain control (AGC) was used; 5E4 ions were cumulated for the formation of MS/MS spectrum. The *m*/*z* scanning ranged from 350 to 1800. The fixed first mass setting is 100 *m*/*z*.

### 4.7. Database Search and Pathway Analysis

The resulting MS/MS data was processed by MaxQuant with an integrated Andromeda search engine (v.1.4.1.2). Search the MS/MS against the SWISS-PROT Sus scrofa database linked with a reverse decoy database. Trypsin/P was designated as a cleavage enzyme, and each peptide could cleave up to 4 deletions, 5 charges, and 5 modifications. The mass error of fragment ions was 0.02 Da and that of precursor ions was 10 ppm. Acetylation on protein N-terminal or Lys was specified as a variable modification, while Cys carbamoyl methylation was specified as an oxidation and fixed modification of Met. The FDR threshold for the peptide, modification and protein site was 1%. The length of minimum peptides was 7. Select the TMT-6plex for the quantification method. Set the site localization probability as >0.75 and set other parameters to default values. KEGG pathway was used for pathway enrichment analysis [47].

### 4.8. Hepatocyte Isolation and Cell Culture 

Isolation of hepatocytes from weaned piglets’ liver was performed by collagenase perfusion and mechanical destruction [48]. Dulbecco’s modified eagle medium (DMEM) with supplement of 10% newborn bovine serum (Biochrom, Berlin, Germany) was used to culture the Chang’s liver cells and HEK293T cells [41]. Chang’s liver cells and HEK293T cells were incubated in accordance with the methods described below. By using trypan blue exclusion, the viability of cells was greater than 90%. In DMEM which was added 10% fetal bovine serum (Sigma, St. Louis, MO, USA), streptomycin (100 μg/mL), and penicillin (100 U/mL) (Invitrogen, San Diego, CA, USA), liver cells were plated at 1.9 × 10^4^ cells/cm^2^ onto glass plates that were collagen-coated. Cells were incubated in a damp environment at 37 °C with 5% CO_2_ for 3 h. Thereafter, in accordance with the protocols, liver cells were incubated and the medium was changed. The cells were washed and sonicated in 0.3 M sucrose (Merck Chemicals, Darmstadt, Germany) at the end [49].

### 4.9. Cells Treatment and Transient Transfection 

As described previously, before harvesting the cells, trichostatin A treatment was performed by adding 10 μM TSA for 16–20 h and nicotinamide treatment was conducted by adding 5–10 mM nicotinamide for 4–8 h [41]. The cDNAs encoding tryptophan 2,3-dioxygenase (TDO) and phenylalanine hydroxylase (PAH) were cloned into Myc, Flag (pcDNA-Myc; pcDNA-Flag). QuickChange Site-Directed Mutagenesis kits (Stratagene, San Diego, CA, USA) were used for producing PAH and TDO point mutations. For Chang’s liver cell, plasmid transfection was conducted by Lipofectamine 2000 (Invitrogen, San Diego, CA, USA), and for HEK293T the calcium phosphate method was used [41]. 

### 4.10. Isotope Tracer Experiments and GC-MS Analysis

The 6-well dishes seeded with hepatocytes was prepared for isotope tracing experiment and then collected as mentioned above. Cells were collected after treatment for 2, 6, 12, and 24 h to determine metabolic homeostasis. Phosphate buffer saline (PBS) was used to wash and scrape cells on ice. Collected media and pellets were stored at −80 °C. The Agilent 7890b GC system, which was connected with an Agilent 5977a mass spectrometer was performed for GC-MS analysis. Helium flow was set to 1 mL/min. The temperature was set as follows: source was maintained at 230 °C, interface at 280 °C, MS quad at 150 °C, and inlet at 250 °C. Mass spectra was recorded with a 4 ms dwell time in selected ion monitoring (SIM) mode. As previously mentioned, ^13^C-labeling of metabolites was ascertained by measuring [50].
(1)$$13C-\mathrm{labeling}\;=\;1/n\sum_{i=1}^{n}\;i\cdot \left(M+i\right)$$
where *n* is the amount of carbon atoms, *M* + *i* is the relative mass isotope abundance after a correction of the natural isotopomer abundance.

### 4.11. Western Blots and Immunoprecipitation

Antibodies which were against PAH, TDO, CRP, HP, Pig-MAP, SAA, mtHSP70, and β-actin (1:1000 dilution, Abcam, Cambridge, UK) were used to detect hepatic proteins [11]. For immunoprecipitation, hepatocyte mitochondrial lysate was mixed with anti-PAH or anti-TAT antibodies (1:1000 dilution, Abcam, Cambridge, UK) at 4 °C overnight and subsequently added to protein A/G beads (Millipore, Burlington, MA, USA) for 4 h. An anti-acetyl-lysine antibody (1:1000 dilution, Abcam, Cambridge, UK) was used for Western blotting. In the NP-40 buffer, which contained a protease inhibitor cocktail (Roche, Basilea, Switzerland), HEK293T cells were lysed. Immunoprecipitation was performed by incubating either FLAG/Myc beads with lysate at 4 °C for 3–4 h or the appropriate antibodies for 2–3 h with lysate, and subsequently incubated Protein-A beads (Upstate). Follow standard Western blot procedure for analysis of tag and proteins. Blocking was accomplished using 50 mM Tris (pH 7.5) with 1% peptone (AMRESCO, Washington, USA) and 10% (*v*/*v*) Tween-20, and primary and secondary antibodies were prepared by using 50 mM Tris (pH 7.5) with 0.1% peptone for acetylation Western blotting [41].

### 4.12. Enzymatic Activity Assay

Culture supernatant of the cells were collected after treatment at specified time points and stored at −80 °C. Homogenize piglets tissue and plasma were put in DMEM and stored at −80 °C. Tissue and plasma sample was uncongealed. After centrifugation, the supernatant was gathered before measurement. The activity of PAH was identified according to the method described by Nielsen [51]. The activity of TDO was identified in accordance with a method that was previously described by Dairam et al. [52].

### 4.13. Statistical Analysis

Statistical analysis was performed by SPSS 20.0 (SPSS, Inc., Chicago, USA). Data were expressed as means ± SDs. Data were analyzed by the standard t-test between 2 groups. Before ANOVA analysis, Levene’s test was performed to assess the unequal variance. Data were analyzed with a significance level of *p* < 0.05 by one-factor ANOVA, and multiple means were compared by using the Tukey–Kramer method. In case of unequal variances, post hoc comparisons were performed by using Welch’s ANOVA and Dunnett’s T3 test.

## Figures and Tables

**Figure 1 metabolites-10-00146-f001:**
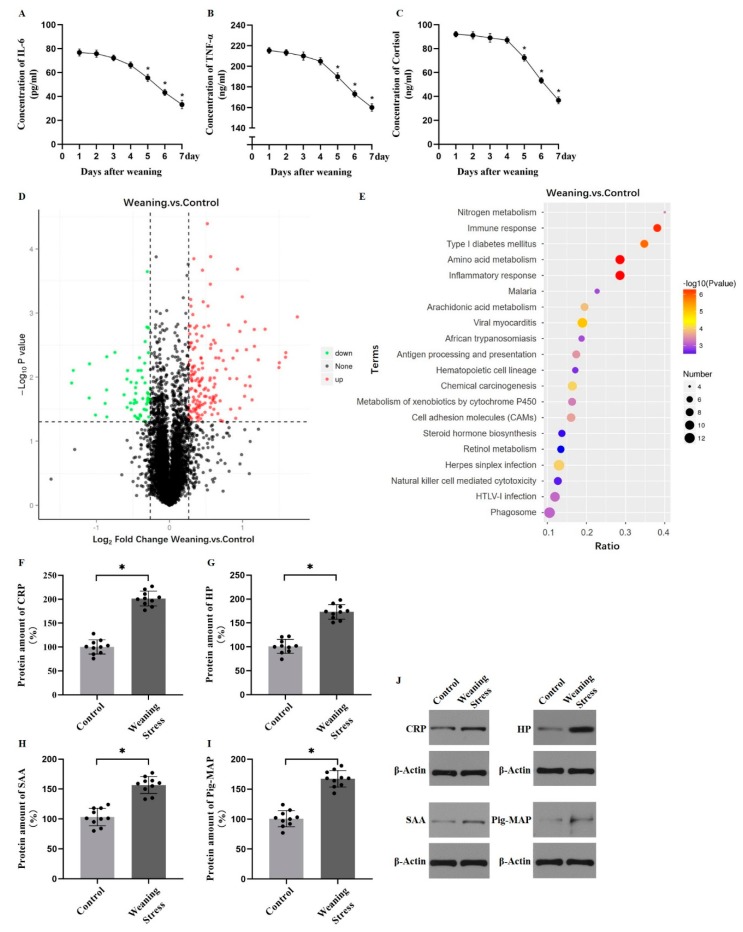
Weaning stress reduces the content of hepatic aromatic amino acids in piglets. (**A**–**C**) Plasma concentration of IL-6 (pg/mL), TNF-α (ng/mL), and cortisol (ng/mL) after weaning at day 1 to day 7. (**D**) Differentially expressed proteins between weaning and control group were shown by volcano plot. Red dots indicate proteins that are up-regulated, and green dots indicate proteins that are down-regulated. (**E**) KEGG analysis between the weaning group and control group. (**F**–**J**) Hepatic C-reactive protein (CRP), haptoglobin (HP), Pig-major acute phase protein (Pig-MAP), and Serum amyloid A (SAA) protein levels in pigs liver. (Data are mean ± SD; *n* = 10 and * *p* < 0.05).

**Figure 2 metabolites-10-00146-f002:**
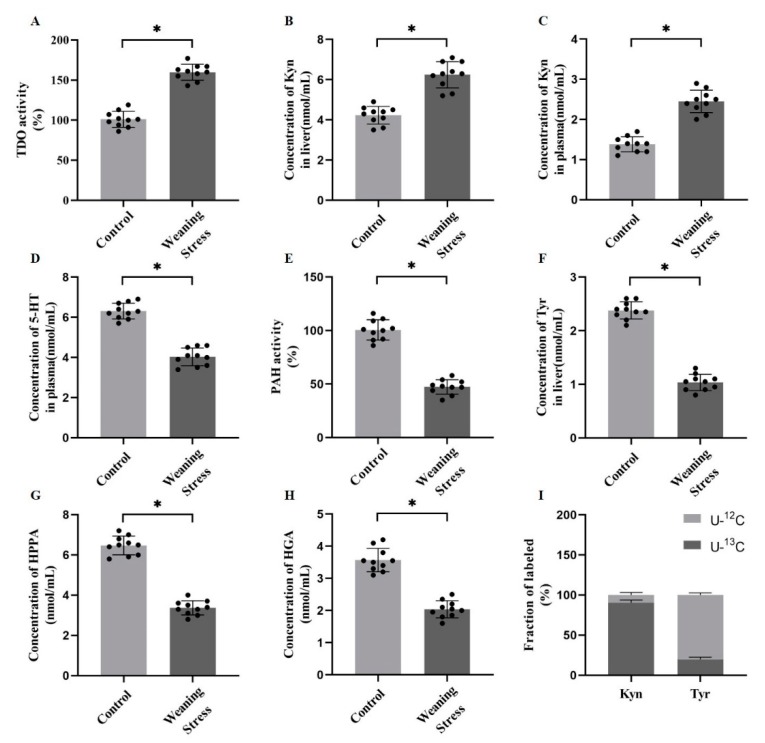
Weaning stress inhibits phenylalanine and tyrosine catabolism and promotes hepatic tryptophan catabolism. (**A**) Tryptophan metabolizes key enzymes, tryptophan 2,3-dioxygenase (TDO) enzyme activity in pig liver. (**B**–**D**) The content of tryptophan metabolites (nmol/mL) after weaning. (**E**) Phenylalanine metabolizes key enzymes, tryptophan 2,3-dioxygenase (PAH) enzyme activity in pig liver. (**F**–**H**) The content of phenylalanine and tyrosine metabolites (nmol/mL) after weaning. (**I**) Kyn and tyrosine labeling pattern from hepatocyte treated with [U-^13^C]-labeled substrates. Error bars represent ± SD, * *p* < 0.05.

**Figure 3 metabolites-10-00146-f003:**
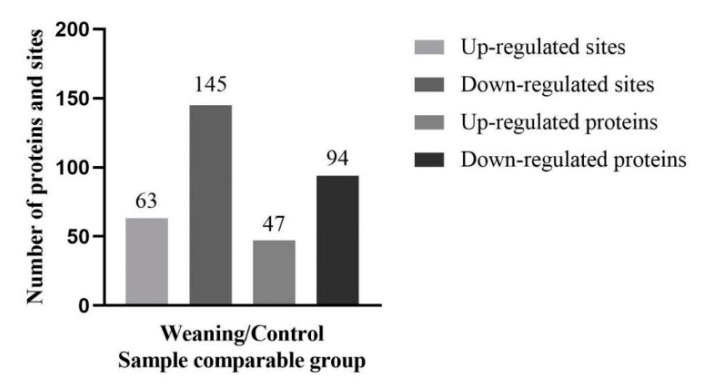
Lysine acetylation modify the key enzyme of hepatic aromatic amino acids (AAAs) metabolism altering the utilization of AAAs. The number of proteins and sites of up-regulated/down-regulated proteins.

**Figure 4 metabolites-10-00146-f004:**
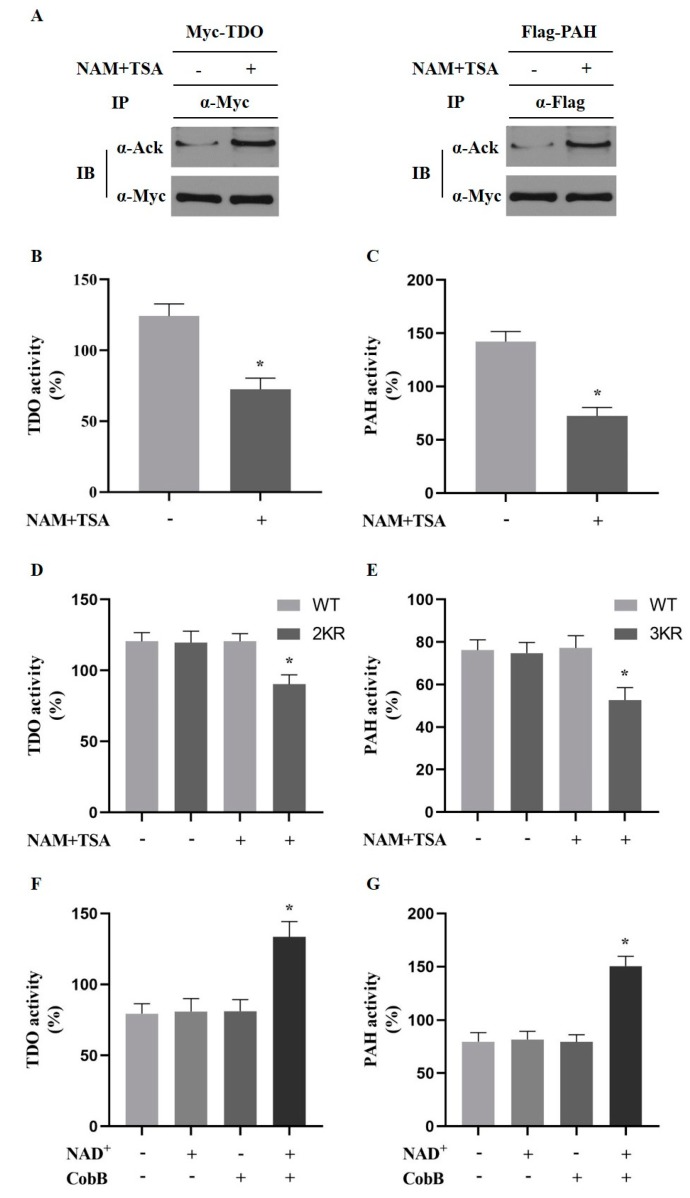
Acetylation of PAH and TDO modulates its activity. (**A**) Acetylation of PAH and TDO. PAH-Myc and Flag-TDO were expressed in HEK293T cells and acetylation were detected by immunoblotting. (**B**–**C**) Hepatic PAH and TDO enzyme activity of weaning piglets compared with control group. (Data are mean ± SD; *n* = 10 and * *p* < 0.05). (**D**–**E**) The activity of PAH and TDO with or without treatment of NAM and TSA in HEK293T cells. KR represents K-to-R mutation; WT represents wild-type (Data are mean ± SD; *n* = 10 and * *p* < 0.05). (**F**–**G**) PAH and TDO enzyme activity with or without CobB deacetylase. As an essential cofactor of CobB, NAD^+^ was omitted as a negative control (Data are mean ± SD; *n* = 10 and * *p* < 0.05).

**Figure 5 metabolites-10-00146-f005:**
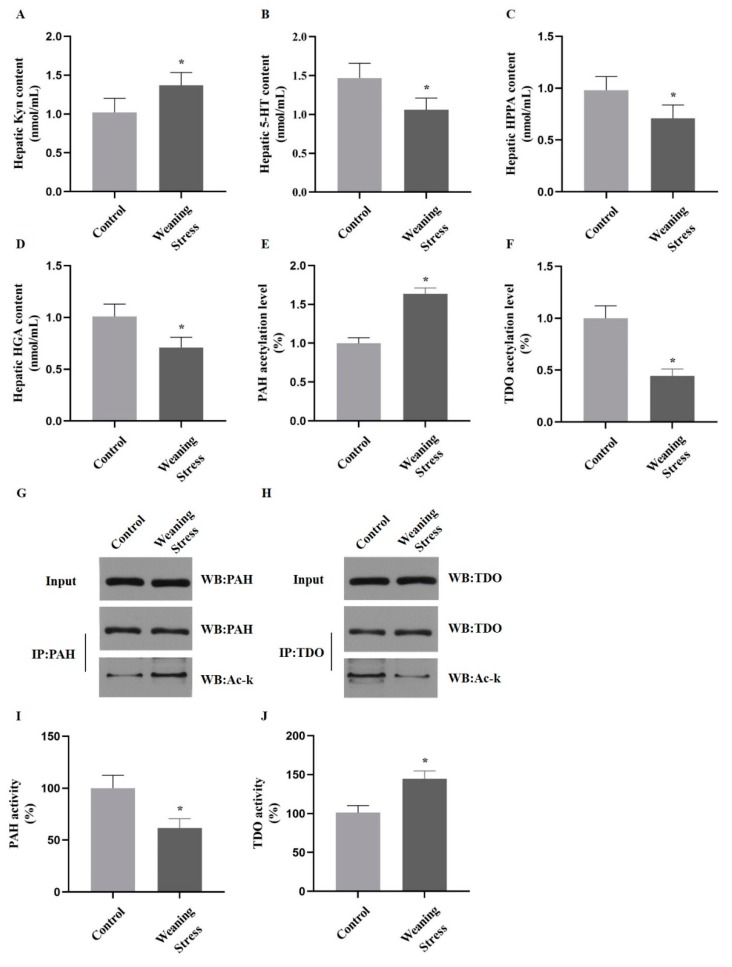
Acetylation of PAH inhibits phenylalanine catabolism and acetylation of TDO promotes tryptophan catabolism in primary hepatocytes. (**A**–**D**) The content of Kyn, 5-HT, HPPA, and HGA (nmol/mL) in primary porcine hepatocytes from liver of weaned piglets. (**E**–**H**) PAH and TDO acetylation levels after weaning compared to control group. (**I,J**) Activity of PAH and TDO in weaning group compared with control group (Data are mean ± SD; *n* = 10 and * *p* < 0.05).

**Table 1 metabolites-10-00146-t001:** Effects of weaning stress on free amino acid content in piglet liver (nmol/mL).

Item	Weaning Group	Control Group	SEM	*p*-Value
Asp	439.56	277.20	78.36	0.106
Glu	2340.47 ^a^	1170.86 ^b^	115.32	0.026
Asn	398.06	343.26	37.39	0.125
Ser	798.58	702.48	55.62	0.132
Cit	15.10	6.250	2.60	0.078
Thr	195.25	102.64	33.40	0.069
Tau	362.53	261.74	71.51	0.083
Ala	2540.13 ^a^	1250.07 ^b^	98.49	0.035
Tyr	651.67 ^a^	891.59 ^b^	56.40	0.046
Trp	161.44 ^a^	361.26 ^b^	36.30	0.062
Met	589.79	779.20	63.39	0.105
Val	1080.71	980.56	154.57	0.074
Phe	677.56 ^a^	913.59 ^b^	93.34	0.039
His	445.62	562.35	112.26	0.076
Ile	596.79	766.27	146.61	0.068
Leu	1210.28	1310.93	182.19	0.078
Orn	654.05	750.79	109.25	0.059
Lys	1100.94	625.02	178.14	0.087
Pro	1110.50	994.83	249.50	0.065

SEM = standard error of mean, *n* = 10; *p*-values were obtained from the analysis of variance (ANOVA) F test; ^a^, ^b^ Means within a row that do not share a same superscript differ (*p* < 0.05).

**Table 2 metabolites-10-00146-t002:** Identification of acetylated phenylalanine hydroxylase (PAH) and tryptophan 2,3-dioxygenase (TDO) sites by mass spectrometry.

Protein Accession	Position	Gene Name	Protein Description	Weaning vs. Control Ratio	Regulated Type	Amino Acid
F1SRJ8	38	PAH	Phenylalanine-4-hydroxylase OS = Homo sapiens	1.53	Up	K
F1SRJ8	204	PAH	Phenylalanine-4-hydroxylase OS = Homo sapiens	1.49	Up	K
F1SRJ8	282	PAH	Phenylalanine-4-hydroxylase OS = Homo sapiens	1.42	Up	K
F1RWA8	94	TDO	Tryptophan 2,3-dioxygenase OS = Homo sapiens	0.68	Down	K
F1RWA8	212	TDO	Tryptophan 2,3-dioxygenase OS = Homo sapiens	0.73	Down	K

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
