# Peer review of "Acetylation of Phenylalanine Hydroxylase and Tryptophan 2,3-Dioxygenase Alters Hepatic Aromatic Amino Acid Metabolism in Weaned Piglets"

_metabolites, 2020, doi:10.3390/metabo10040146_

Round 1
Reviewer 1 Report
I enclose the PDF which includes MANY comments, that I hope you and the authors can see. In general this paper makes a valuable contribution in showing that acetylation of two key enzymes in aromatic amino acid metabolism may provide a mechanism for regulating enzyme function, and that this may be part of the stress response associated with weaning. Unfortunately parts of the writing are unclear, and I have tried to mark those sections. Most significantly the authors fail to clearly explain the relationship between what they are measuring and aromatic amino acid metabolism. I have marked these sections. It is insufficient to say there is a relationship without explaining the relationship and providing appropriate references. Although the manuscript remains premature, I would be happy to evaluate a revision.
Reviewer 2 Report
The manuscript entitled “Acetylation of phenylalanine hydroxylase and tryptophan 2,3-dioxygenase alters hepatic aromatic amino acid metabolism in weaned piglets” provides novel knowledge that may contribute to reveal the mechanism of alteration of hepatic aromatic amino acid metabolism via weaning. I read the manuscript with certain interest, and I think that it may be also had interest by many researchers. However, there are concerns in this manuscript as below.
Concerns:
- Abbreviations
Some abbreviations (e.g. NAM, TSA, Ac-k, and APP) are used without definition. In addition, IL-6 and TNF-a are used in "Results" (p. 2) while these are defined in "Methods and materials" (p. 11).
- Tables
I cannot find Table 1 and 2 in the manuscript.
- p. 2, lines 81~83, p.8, lines 241~243
I think that the explanation by the authors should be insufficient. For example, did the weaned piglets develop phenylketonuria? I think that the authors should investigate alteration of the concentrations of phenylpyruvate, phenylacetate and phenyllactate in the weaning piglets.
- Figure 4A~E
Cells were treated with TSA and NAM. How about the effect of only TSA?
- Figure 4F and G
Are the descriptions (NAD+: - - + +; CobB: - - + +) under the graphs correct?
e.g. NAD+: - - + +; CobB: - - + + → NAD+: - + - +; CobB: - - + +)
- p.6, line 169
acetylation → deacetylation
- Figure 5
Primary hepatocytes were isolated and cultured in the experiments. How about the effects of isolation of the cells from piglets on the state of the cells? For example, the epigenetic state of the hepatocytes may change by the stress caused by isolation from piglets.
In p.7 line 191~192, the authors described "P<0.05, Fig. 5E-F". Did the authors carry out quantitative analysis of the acetylation levels of PAH and TDO (Fig. 5E and F)? I think that the authors should show the results of quantitative analysis of the acetylation levels of PAH and TDO as graphs.
In addition, I think that "Treatment" should be changed to "Weaning group" (see p.7, line 186) in Fig. 5A~D, G and H.
- p.10, lines 315~317
The authors used nicotinamide (NAM) in this study (Fig. 4). Do the authors think the sirtuins as the probable candidates? How about describing sirtuins as the candidates of the responsible deacetylases for AAA metabolism in "Discussion" section?
Round 2
Reviewer 1 Report
The authors have sufficiently altered the manuscript with regard to word choice and clarity. The rationale for the study (why the chosen experiments reflect stress) remains unclear. What the authors have added to rationalize the chosen experiments consists of statements of belief, not proven science with appropriate references. Nevertheless, if these are common beliefs in the field, the manuscript is acceptable for publication.